# Dopamine promotes instrumental motivation, but reduces reward-related vigour

**John P Grogan[1]\*, Timothy R Sandhu[1,2], Michele T Hu[1,3], Sanjay G Manohar[1,4]**

[1]Nuffield Department of Clinical Neurosciences, University of Oxford, Oxford, United Kingdom; [2]Department of Psychology, University of Cambridge, Cambridge, United Kingdom; [3]Oxford Parkinson's Disease Centre, University of Oxford, Oxford, United Kingdom; [4]Department of Experimental Psychology, University of Oxford, Oxford, United Kingdom

**Abstract** We can be motivated when reward depends on performance, or merely by the prospect of a guaranteed reward. Performance-dependent (contingent) reward is instrumental, relying on an internal action-outcome model, whereas motivation by guaranteed reward may minimise opportunity cost in reward-rich environments. Competing theories propose that each type of motivation should be dependent on dopaminergic activity. We contrasted these two types of motivation with a rewarded saccade task, in patients with Parkinson's disease (PD). When PD patients were ON dopamine, they had greater response vigour (peak saccadic velocity residuals) for *contingent* rewards, whereas when PD patients were OFF medication, they had greater vigour for *guaranteed* rewards. These results support the view that reward expectation and contingency drive distinct motivational processes, and can be dissociated by manipulating dopaminergic activity. We posit that dopamine promotes goal-directed motivation, but dampens reward-driven vigour, contradictory to the prediction that increased tonic dopamine amplifies reward expectation.

**\*For correspondence:**
john.grogan@ndcn.ox.ac.uk

## Introduction

Organisms expend more effort when their actions can lead to rewards, as the value of the reward offsets the extra effort expended to attain them (*Kool and Botvinick, 2018*; *Manohar et al., 2015*; *Niv et al., 2006*; *Shenhav et al., 2017*). They will even do so if the extra effort does not increase the reward they receive (*Glaser et al., 2016*; *Milstein and Dorris, 2007*; *Xu-Wilson et al., 2009*), indicating that mere expectation of reward is enough to justify the effort cost. Motivation, which promotes this effort expenditure, has two facets: it allows actions to be directed towards goals, and it energises our actions when rewards are expected (*Niv et al., 2006*). These two aspects are not always coupled. For example, employees might be salaried, where a fixed reward is *guaranteed* irrespective of achievements, or they might receive merit-based pay that is *contingent* on meeting performance targets (*Lazear, 2000*).

Contingent rewards motivate us because we understand the causal relation between successful actions and reward. This is instrumental, in that we apply knowledge of action-outcome associations. For instance, people must realise that merit-based pay depends on their performance for it to incentivise them. In animals, dopaminergic input to dorsal striatum is necessary for instrumental motivation (*Lex and Hauber, 2010b*).

In contrast, reward that is independent of what an agent does might motivate us because in a variable environment, we capitalise on rewards while they are available (*Niv et al., 2007*). One proposed mechanism for this is that tonic dopamine encodes expected reward rate, such that in a rich environment agents are motivated to respond faster to maximise the rewards they receive

(*Niv et al., 2007*). Equally, dopamine can be viewed as signalling an opportunity cost– time is more costly when reward is available, and so organisms act faster (*Otto and Daw, 2019*; *Shadmehr et al., 2010*). The dopaminergic drive has not only generalised motivating effects, termed vigour (*Beierholm et al., 2013*; *Guitart-Masip et al., 2011*; *Niv et al., 2007*), but also context-specific effects. For example, a stimulus that predicts rewards drives conditioned responses that are uncoupled with reward (*Lovibond, 1981*) – similar to how salary increases might improve job performance. This phenomenon, known as Pavlovian-to-Instrumental transfer, requires dopamine projections to nucleus accumbens (*Hall et al., 2001*; *Kelley and Delfs, 1991*; *Talmi et al., 2008*; *Wassum et al., 2013*; *Wyvell and Berridge, 2000*). Similarly, animals tend to approach stimuli associated with rewards, even in the absence of action-contingency, a behaviour called autoshaping or sign-tracking, which also relies on nucleus accumbens dopamine (*Day et al., 2006*; *Di Ciano et al., 2001*).

The dopaminergic basis of instrumental and Pavlovian motivation could potentially explain the impaired motivation seen in PD patients and the rescue of such deficits by rewards (*Ang et al., 2018*; *Chong et al., 2015*; *de Wit et al., 2011*; *Kojovic et al., 2014*). However in certain situations, motivation by reward can paradoxically be *stronger* in patients with low dopamine (*Aarts et al., 2012*; *Timmer et al., 2018*), making dopamine's exact role in motivation unclear.

These two effects of contingent and expected rewards frequently overlap in real life and in previous experiments – higher stakes raise reward expectation, but also mean that actions carry more weight. However, experimental control of expectation and contingency allows them to be dissociated (*Manohar et al., 2017*), which reveals that both contingency and expectation can separately motivate behaviour, and that these effects are independent rather than correlated or antagonistic, suggesting distinct mechanisms.

We used this incentivised saccade task (*Manohar et al., 2017*) here to test PD patients ON and OFF their dopaminergic medication, along with healthy age-matched controls. We tested the two predictions that dopamine is involved in motivation by expected rewards, and by contingent rewards.

## Results

### Dopamine promotes contingent motivation and attenuates reward-expectation motivation

Participants made saccades to a target after hearing cues indicating how reward would be determined (*Figure 1b*). To measure motivation by contingent rewards, we compared trials where rewards were delivered depending on participants' response times (Performance), to trials where rewards were given with 50% probability (Random). We matched the average reward rate so that both these conditions had identical reward expectation and uncertainty, and only differed in their *contingency*. To measure motivation by reward expectation, we compared trials with a guaranteed reward (10 p) to those with a guaranteed no-reward (0 p). In both these conditions rewards were delivered unconditionally, and only differed in terms of *expected reward*. We tested 26 PD patients ON and OFF dopaminergic medication (PD ON and PD OFF) and 29 healthy age-matched controls (HC) on a rewarded eye-movement task that separated effects of contingent and non-contingent motivation (see *Figure 1a* for task, see *Table 1* for participant details).In all trials, feedback was given about whether the response was fast or slow, in addition to the reward received, to control for intrinsic motivation. A saccade's velocity is tightly governed by its amplitude, a relation known as the 'main sequence' (*Bahill et al., 1975*). To account for this, we regressed out the effect of amplitude on peak velocity leaving peak saccade velocity residuals as our main measure of response vigour (see *Figure 1e*), as in previous work (*Blundell et al., 2018*; *Manohar et al., 2017*; *Muhammed et al., 2020*; *Muhammed et al., 2016*; *Van Opstal et al., 1990*). This measures how much faster each saccade is than the speed predicted from its amplitude. Thus, positive (negative) residuals mean a particular saccade was faster (slower) than predicted by the main sequence, and makes response vigour independent of any changes to saccade amplitude also caused by our manipulations or by group differences between PD patients and HC. We did this regression for each participant and session separately, but across conditions. A three-way repeated-measures ANOVA

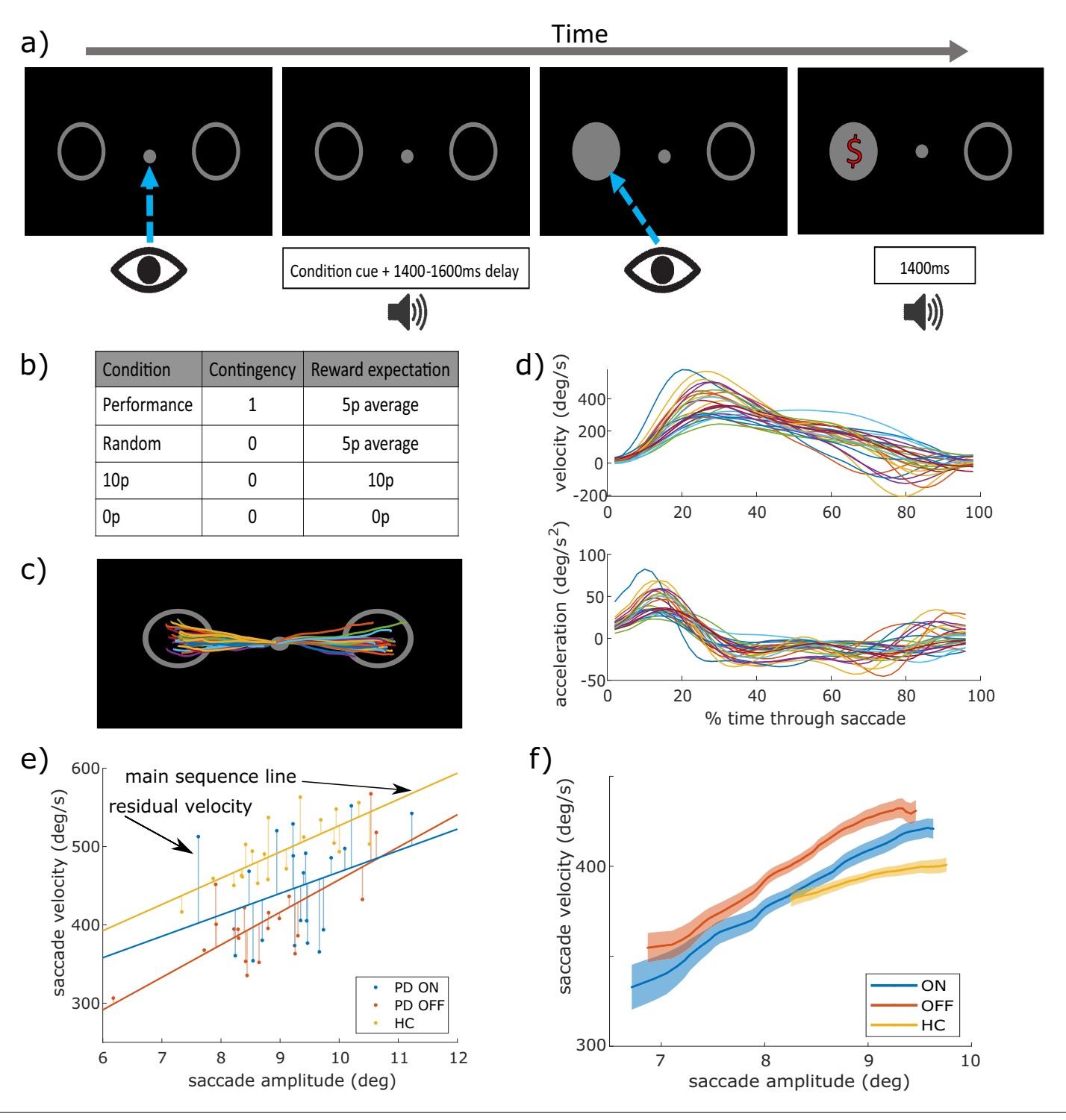

**Figure 1.** Saccade task design and example eye-tracking traces. (**a**) Trial design: participants fixated on the centre, heard a cue for the condition (Performance/Random/10 p/0 p), waited a delay (1400/1500/1600 ms) and then looked towards to the circle that lit up, and were given 10 p or 0 p reward depending on the condition, along with feedback on their response time (fast/slow). (**b**) To measure contingent motivation, we compared 'Performance' trials, where participants had to be faster than their median RT to win reward (thus giving 50% trials rewarded on average), with 'Random' trials where a random 50% of trials were rewarded. To measure motivation by expected reward we compared '10 p' trials where rewards were guaranteed, with '0 p' trials where no-reward was guaranteed. (**c**) Example eye-position traces for one participant and condition (different colours are different trials). (**d**) Example mean velocity and acceleration profiles for all PD ON in the 10 p condition. (**e**) Example of the main sequence and velocity residuals – the points show a subset of individual trials illustrating the 'main sequence' relationship where larger saccades have greater velocity, shown

*Figure 1 continued on next page*

*Figure 1 continued*

by the regression line. The distance from each point to its line is the velocity residual, which we take as out main measure of response vigour. (**f**) Peak velocity of individual saccades increases with the amplitude of movement – the 'main sequence'; example showing the 10 p condition, for PD ON, OFF and HC. Saccadic vigour, our measure of interest, was indexed by the residuals after regressing out amplitude from peak velocity, for each participant.

tested whether dopamine differentially affected contingent and guaranteed motivation – manifest by a three-way (contingency*motivation*drug) interaction.

Dopaminergic medication significantly modulated how contingent and guaranteed motivation affected motor vigour (*Figure 2a*, three-way interaction on peak velocity residuals, p=0.0023; see *Table 2* for statistics). This was because, when ON medication, patients were motivated by contingency but not reward expectation (separate two-way ANOVA in PD ON: contingency*motivation, p=0.0170; see *Supplementary file 1A*), whereas after overnight withdrawal of medication there was a borderline significant interaction in the opposite direction, as PD OFF were motivated by reward expectation but not contingency (PD OFF ANOVA: p=0.0501; *Supplementary file 1A*). This indicates that when PD patients were ON medication, motivation was strongest when reward was contingent on performance, but when they were OFF medication, patients were motivated by guaranteed rewards.

To confirm that the effects on peak velocity residuals were not driven by changes in other aspects of saccades, the same 3-way ANOVA was run on each of the other saccade measures. There were no significant effects on saccadic amplitude (see *Table 2* and *Figure 2c*). Saccadic RT had an effect of contingency as saccades started faster for Performance and Random conditions than 10 p or 0 p conditions (*Figure 2d*, p=0.0396). Endpoint variability had a contingency*motivation interaction (*Figure 2e*, p=0.0482) as variability was higher for 0 p condition. Raw peak velocity had an effect of motivation, as both types of motivation increased speed (*Figure 2f*, p=0.0110), although this will include effects of changes in amplitude (via the main sequence) which showed a borderline significant effect of motivation (*Figure 2c*, p=0.0607).

**Table 1.** Participant demographics for PD patients and Healthy Controls (HC) included in the analysis.

Standard deviations are given in parentheses. **=p < 0.01 (independent samples t-test). ACE = Addenbrooke's Cognitive Exam, AMI = Apathy and Motivation Index, HADS = Hospital Anxiety and Depression scores (A and D given separately), BDI-II = Beck Depression Inventory-II, FSS = Fatigue Severity Scale, UPDRS-III = Unified Parkinson's disease rating scale Part 3, performed ON and OFF medication, LED = Daily Levodopa Equivalent Dose, # on agonists = number of patients taking dopamine agonists in addition to levodopa.

|  | PD | HC |
| --- | --- | --- |
| Number | 26 | 29 |
| Age | 67.69 (1.48) | 67.41 (6.83) |
| Gender (M:F) | 19:7 | 15:14 |
| ACE | 93.04 (6.47) | 97.10 (2.11)** |
| AMI | 1.48 (0.56) | 1.28 (0.47) |
| HADS-A | 2.92 (2.92) | 4.29 (2.79) |
| HADS-D | 2.50 (1.84) | 2.17 (1.83) |
| BDI-II | 4.90 (3.60) | 5.84 (3.78) |
| FSS | 3.19 (1.21) | 3.02 (1.03) |
| UPDRS-III ON | 26.69 (9.20) | N/A |
| UPDRS-III OFF | 35.04 (11.17) | N/A |
| LED | 490.23 (324.28) | N/A |
| # on agonists | 6 | N/A |

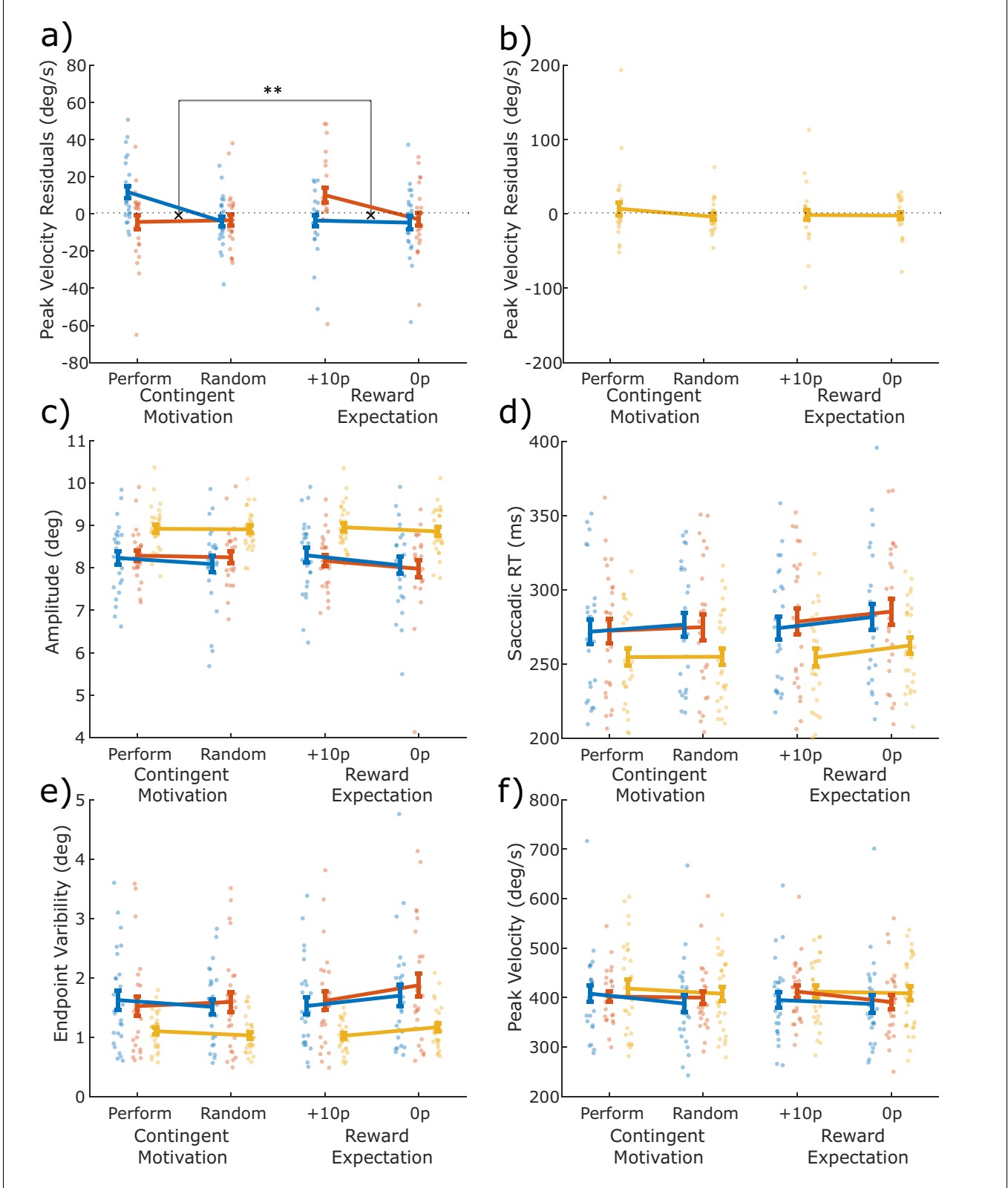

**Figure 2.** Differential effects of dopamine on two types of motivation. The mean measures for the four conditions (Performance, Random, Guaranteed 10p, Guaranteed 0p) for each variable, with individual data points. The difference between Performance and Random shows the effect of Contingent motivation, while the difference between 10p and 0p shows the motivating effect of reward expectation. (a) Peak velocity residuals indexed behavioural vigour. When ON dopamine, patients were motivated to invigorate their saccades when reward depended on response time, but not when expecting a

*Figure 2 continued on next page*

*Figure 2 continued*

guaranteed reward. In contrast, when OFF dopamine, vigour was driven by expectation of guaranteed reward, but not by contingency (F (1, 200) = 9.5190, p = .0023, $\eta_p^2$ = . 0454). (b) HC were similar to PD ON dopamine (please note the different y-axis limits). (c–e) No dopaminergic effects were observed for (c) saccade amplitude, (d) saccade RT, (e) endpoint variability, or (f) raw peak velocity, although PD patients had slower, smaller and more variable saccades than HC. All measures are in visual degrees, except saccade RT (ms). Error bars show within-subject SEM. Statistics are presented in *Table 2*. Data are available in *Figure 2—source data 1*.

The online version of this article includes the following source data for figure 2:

**Source data 1.** Source individual data for all saccade measures for PD ON, OFF and HC.

The HC peak velocity residuals were not affected by contingency, motivation or the interaction (p>0.05; see *Table 3*), suggesting that healthy older adults do not adjust their response vigour for contingent or guaranteed rewards. There were also no significant effects on amplitude, saccadic RT, or raw peak velocity in HC, although endpoint variability did have a significant contingency*motivation interaction (p=0.0048; see *Table 3*). Post-hoc pairwise comparisons showed this was due to guaranteed rewards significantly reducing variability (p=0.0316), while contingent rewards did not (p=0.1219).

We also compared both PD ON and OFF separately against the HC with three-way mixed ANOVA, to see under which conditions patients deviated from healthy behaviour. As expected, HC had overall larger amplitudes, quicker saccadic RTs and lower endpoint variability than both PD ON or OFF (*Figure 2*, see *Supplementary file 1B-C* for statistics). The use of peak velocity residuals rather than raw velocity factors out the effects of PD on movement amplitude, allowing comparison of the motivational changes in velocity while controlling for differences in the main sequence (*Bahill et al., 1975*; *Manohar et al., 2017*). HC did not significantly differ from PD ON or OFF in peak velocity residuals, although their pattern was numerically closest to PD ON with greater contingent motivation.

We additionally checked whether there were practice effects in the PD patients, in case patients behaved differently on their second session due to different expectations. We found no effects or interactions of session on any measure in PD patients (p>0.05).

## Velocity profiles

The effects above demonstrate peak velocity shows strong effects of reward and dopamine, so next we examined the time-course of how velocity was modulated during a saccade. We computed the velocity across time within the movements, and compared the reward effects for PD ON and OFF using cluster-wise permutation tests. Contingent rewards (Performance – Random) did not significantly affect velocity or acceleration for PD ON or OFF, as permutation tests for each condition and the difference between conditions found no significant clusters (cluster-wise permutation tests: p>0.05; *Figure 3a&b*). However, guaranteed rewards (10 p – 0 p) lead to greater velocity early in the saccade for PD OFF (p<0.05; *Figure 3c*), which was significantly different from PD ON (p<0.05). Acceleration traces showed this was due to PD OFF having greater acceleration early in the movement (*Figure 3d*, p<0.05). HC showed no effects of contingent or guaranteed rewards on velocity or acceleration, perhaps unsurprising as there were no differences in overall velocity as reported above. Permutation testing revealed no differences between HC and PD ON or OFF for velocity or acceleration (p>0.05).

Faster movements are known to be more error-prone (*Harris and Wolpert, 1998*; *Harris and Wolpert, 2006*), but motivation can attenuate this effect, making movements more accurate (*Manohar et al., 2019*). Autocorrelation of eye position over time within saccades provides an indicator of corrective motor signals during movements: noise accumulates during movements, so that variability early in a movement causes endpoint error. This is manifest in autocorrelation, where across trials the eye position at early time-points predicts late time-points. Negative feedback signals correct movement errors during the saccade, and manifest as reductions in this autocorrelation (*Codol et al., 2020*; *Manohar et al., 2019*). This feedback, provided by corrective motor signals, can be increased by incentives (*Codol et al., 2019*; *Manohar et al., 2019*). In the current study, guaranteed rewards led to greater autocorrelation early in the saccades for PD OFF than ON (*Figure 4e & g*). This coincides with the greater acceleration PD OFF patients had at the beginning

**Table 2.** Statistics for main behavioural analyses.

Three-way (motivation*contingency*drug) repeated-measures ANOVA on each behavioural measure, for the PD patients ON and OFF medication. An effect of contingency means the guaranteed conditions (10 p, 0 p) were different to the contingent conditions (Performance, Random). An effect of motivation means the 10 p and Performance conditions were different to the Random and 0 p conditions. An interaction of the two means that contingent rewards differed from guaranteed rewards. The Contingency * Motivation * Drug condition means that the effects of contingent and non-contingent rewards differed by PD medication state. Significant effects are highlighted in red. *p < 0.05, **p < 0.01.

| Measure | Effect | F (1, 200) | p | $\eta_p^2$ |
|---|---|---|---|---|
| Peak Velocity Residuals | Motivation | 9.7704 | *.0020 | .0466 |
| | Contingency | 0.0194 | .8895 | .0001 |
| | Drug | 0.0004 | .9850 | .0000 |
| | Motivation * Contingency | 0.0051 | .9429 | .0000 |
| | Motivation * Drug | 0.2626 | .6089 | .0013 |
| | Contingency * Drug | 11.1072 | **.0010 | .0526 |
| | Contingency * Motivation * Drug | 9.5190 | **.0023 | .0454 |
| Amplitude | Motivation | 3.5577 | .0607 | .0175 |
| | Contingency | 1.2284 | .2690 | .0061 |
| | Drug | 0.0000 | .9984 | .0000 |
| | Motivation * Contingency | 0.5545 | .4573 | .0028 |
| | Motivation * Drug | 0.2278 | .6337 | .0011 |
| | Contingency * Drug | 1.7763 | .1841 | .0088 |
| | Contingency * Motivation * Drug | 0.0287 | .8655 | .0001 |
| Saccadic RT | Motivation | 3.4333 | .0654 | .0169 |
| | Contingency | 4.2922 | *.0396 | .0210 |
| | Drug | 0.3560 | .5514 | .0018 |
| | Motivation * Contingency | 0.3663 | .5457 | .0018 |
| | Motivation * Drug | 0.0694 | .7925 | .0003 |
| | Contingency * Drug | 0.6246 | .4303 | .0031 |
| | Contingency * Motivation * Drug | 0.0185 | .8920 | .0001 |
| Endpoint Variability | Motivation | 2.6780 | .1033 | .0132 |
| | Contingency | 3.6181 | .0586 | .0178 |
| | Drug | 1.0095 | .3162 | .0050 |
| | Motivation * Contingency | 3.9524 | *.0482 | .0194 |
| | Motivation * Drug | 1.2787 | .2595 | .0064 |
| | Contingency * Drug | 1.3819 | .2412 | .0069 |
| | Contingency * Motivation * Drug | 0.1626 | .6872 | .0008 |
| Raw Peak Velocity | Motivation | 6.5921 | *.0110 | .0319 |
| | Contingency | 0.3831 | .5366 | .0019 |
| | Drug | 1.8937 | .1703 | .0094 |
| | Motivation * Contingency | 0.1179 | .7316 | .0006 |
| | Motivation * Drug | 0.0563 | .8126 | .0003 |
| | Contingency * Drug | 0.5462 | .4608 | .0027 |
| | Contingency * Motivation * Drug | 2.4061 | .1224 | .0119 |

**Table 3.** Statistics for behavioural analysis on HC saccade data.
HC had a motivation*contingency interaction for endpoint variability, as only expected rewards decreased variability. **=p < 0.01.

| Group | Effect | F (df = 1, 112) | p | $\eta_p^2$ |
|---|---|---|---|---|
| Peak Velocity Residuals | Motivation | 0.9019 | .3443 | .0080 |
| | Contingency | 0.3463 | .5574 | .0031 |
| | Motivation * Contingency | 0.6995 | .4047 | .0062 |
| Amplitude | Motivation | 2.3510 | .1280 | .0206 |
| | Contingency | 0.0255 | .8734 | .0002 |
| | Motivation * Contingency | 1.2551 | .2650 | .0111 |
| Saccade RT | Motivation | 3.2227 | .0753 | .0280 |
| | Contingency | 2.5743 | .1114 | .0225 |
| | Motivation * Contingency | 2.7992 | .0971 | .0244 |
| Endpoint Variability | Motivation | 0.9304 | .3368 | .0082 |
| | Contingency | 0.6651 | .4165 | .0059 |
| | Motivation * Contingency | 8.2781 | **.0048 | .0688 |
| Raw Peak Velocity | Motivation | 1.1321 | .2896 | .0100 |
| | Contingency | 0.1615 | .6885 | .0014 |
| | Motivation * Contingency | 0.2538 | .6154 | .0023 |

of saccades to guaranteed rewards (*Figure 3d*), as faster movements have greater motor noise (*Harris and Wolpert, 1998*; *Harris and Wolpert, 2006*). Notably, this reward-related autocorrelation did not persist until the end of the saccade, suggesting that negative feedback corrected it. However, as we did not find decreased autocorrelation around the end of the saccades, this represents only indirect evidence of negative feedback.

## No correlation of the velocity effects for the distinct motivational processes

Previous work had shown that motivation by contingent and guaranteed reward did not correlate across participants (*Manohar et al., 2017*), so we asked whether dopamine's effects upon these two types of motivation was also uncorrelated. We found no correlation between effects of contingent and guaranteed rewards on peak saccade velocity residuals in PD ON, PD OFF or HC separately, nor a correlation between medication states, nor between the drug-induced changes in the effects (p>0.05; see *Figure 5* legend for statistics). This suggests that the two effects are separate and independent, and not antagonistic within the same person. In particular, the degree to which dopamine improved performance-contingent motivation did not predict the degree to which it reduced motivation by guaranteed rewards.

Source data are available in *Figure 5—source data 1*.

## Pupil dilatation

We examined pupil dilatation after the cue onset and before the target appeared (after 1400 ms). Previous research has shown a greater effect of contingent than guaranteed reward on pupil dilatation, maximal around 1200 ms after the cue (*Manohar et al., 2017*), so we used a window-of-interest analysis on the mean pupil dilatation 1000–1400 ms after the cue. There were no significant effects or interactions (p>0.05; *Figure 6*, see *Supplementary file 2A-C* for statistics), suggesting that dopamine and reward did not affect pupil responses in PD patients.

We also used a hypothesis-free analysis, using cluster-wise permutation testing across the whole time-course to look for significant differences between conditions and groups, which also found no significant effects (p>0.05).

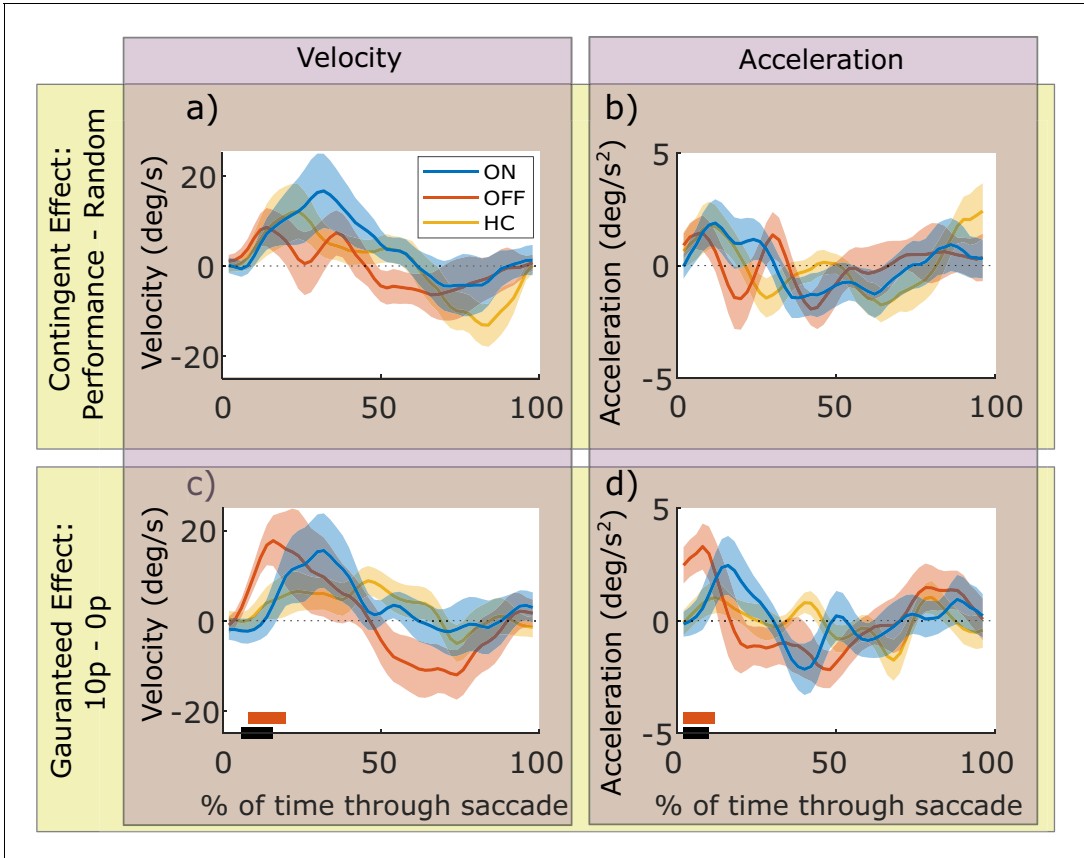

**Figure 3.** Motivational effects on instantaneous velocity and acceleration within a saccade. The top row shows the effects of contingent rewards (i.e. measures in Performance conditions minus the Random condition), and the bottom row shows effects of guaranteed rewards (10 p condition minus 0 p condition). The x-axis is % of normalised time where 0 indicates the start of a saccade, and 100 is the end. The instantaneous velocity (a and c) is increased by contingent (a) and guaranteed (c) rewards, and PD patients OFF have an earlier and greater increase in velocity for guaranteed rewards than PD ON. The orange bar shows time-points where PD OFF had velocity significantly greater than zero (cluster-wise permutation tests, p<0.05), the black bar shows time-points where PD ON and OFF significantly differed (PD ON and HC did not differ from zero, so there are no blue or yellow bars). Acceleration traces (b and d) showed this was due to guaranteed motivation increasing acceleration at the start of the movement for PD OFF (d; significant cluster, p<0.05). Shading shows SEM. Source data are available in *Figure 3—source data 1*. *Figure 3—figure supplement 1*. Individual participants' velocity and acceleration traces.

The online version of this article includes the following source data and figure supplement(s) for figure 3:

**Source data 1.** Source individual data for saccade velocity and acceleration for PD ON, OFF and HC.

**Figure supplement 1.** Individual participants' velocity and acceleration traces.

We found no correlations between pupil dilatation and motivation effects in any group, or overall (p>0.05; *Figure 6—figure supplement 1*). Thus, the vigour effects were not related to pupillary dilatation before the movement.

## PD severity

We looked to see whether the dopaminergic effects on velocity residuals could be tied to PD symptom expression. The UPDRS (*Martínez-Martín et al., 2015*) is a measure of PD symptom severity and was performed in each session; part III measures motor symptom severity. We found no correlations between UPDRS-III scores and reward effects on peak velocity residuals in PD ON (Guaranteed: ρ = −0.1256, p=0.5410; Contingent: ρ = −0.2327, p=0.2527) or OFF (Guaranteed: ρ = −0.2067, p=0.3110; Contingent: ρ = 0.1553, p=0.4487). Thus, the reward effects were unrelated to PD symptom severity.

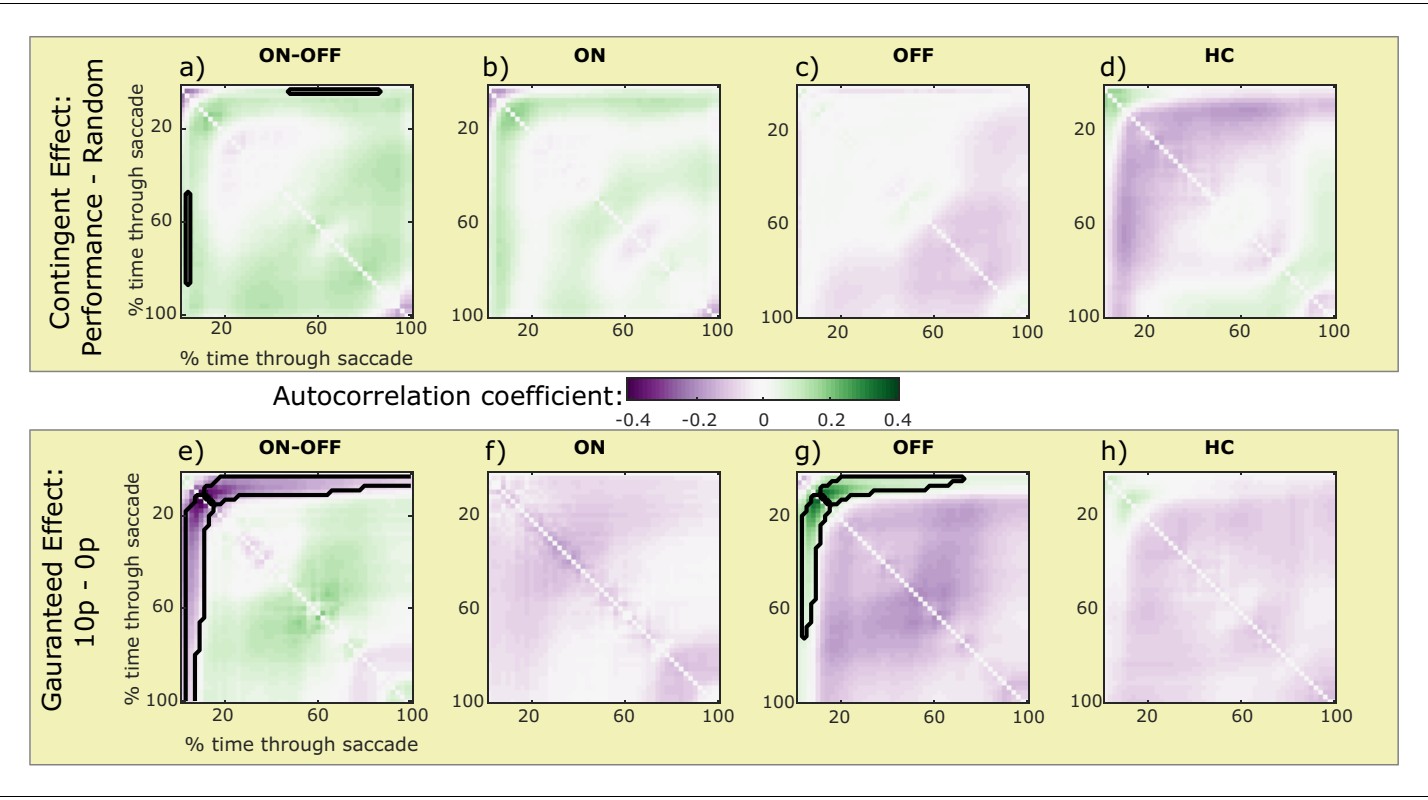

**Figure 4.** Motivational effects on eye-position autocorrelation within saccades. Each image shows the effect of reward on mean correlation coefficient between the eye-position at one (interpolated) time-point within a saccade with all other time-points in that same saccade. As noise accumulates during the movements, the correlations increase over the time-points, while reductions in correlation can suggest negative feedback during movements. The top row shows the effect of contingent rewards (Performance – Random) on the (Fisher transformed) autocorrelation coefficients, and the bottom row shows the effect of guaranteed rewards (10 p – 0 p). Green areas mean that motivation increased correlation, while purple areas reflect a decrease, and clusters significantly different from zero are outlined in black (cluster permutation testing, p<0.05). When examining the dopaminergic effects (a and e: PD ON – OFF), a significant cluster was found, such that patients differed in their correlations early in the saccade when rewards were guaranteed (e). This was due to guaranteed rewards increasing early correlation only for PD OFF (g). The time of this increase matches the time of increased acceleration shown in *Figure 3d*. There was also a small cluster of significant difference between PD ON and OFF for contingent rewards (a), but there were no clusters within ON (b) or OFF (c) separately. HC had no clusters of significant differences (d and h). Source data are available in *Figure 4—source data 1*. *Figure 4—source data 2*. Individual data for autocorrelation. *Figure 4—figure supplement 1*. Motivational effects on saccade time-time covariance within saccades.

The online version of this article includes the following source data and figure supplement(s) for figure 4:

**Source data 1.** Source individual data for autocorrelation coefficients for PD ON, OFF and HC.
**Source data 2.** Individual participants' autocorrelation matrices.
**Figure supplement 1.** Motivational effects on time-time covariance within saccades.

## Depression and apathy

We gave participants questionnaires measuring apathy, the AMI (*Ang et al., 2017*) and depression, BDI-II and HADS (*Beck et al., 1996*; *Zigmond and Snaith, 1983*). We found no significant correlations between these questionnaires and contingent or guaranteed motivational effects on peak velocity residuals in PD ON or OFF (p>0.05, see *Supplementary file 4* for statistics).

## Fixation period

We looked at whether motivation was affecting behaviour during the fixation period (1400 ms between condition cue onset and target onset) differently, which could potentially lead to differences during the movements. We excluded trials with saccades, blinks, deviations greater than 1.8° and segments with velocities greater than $30°s^{-1}$.

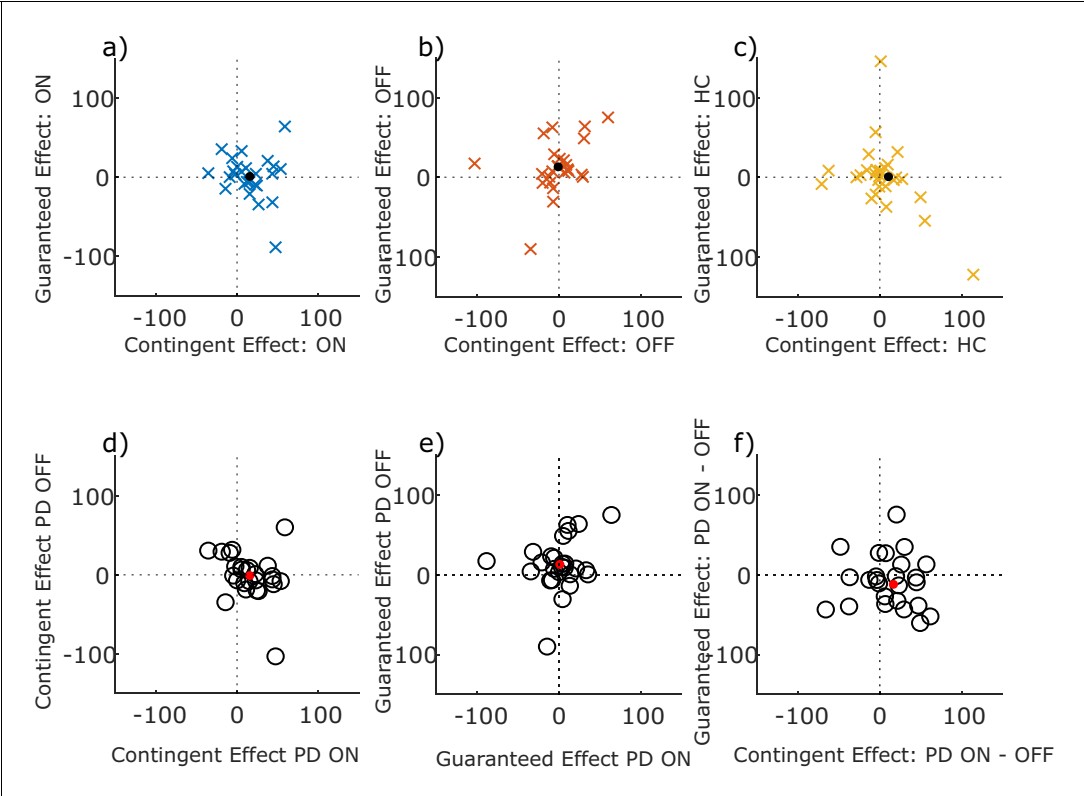

**Figure 5.** No correlations between contingent and guaranteed rewards. Scatter plots of the effect of contingent and guaranteed rewards (i.e. contingent effect = Performance minus Random trials, guaranteed effect = guaranteed 10 p minus guaranteed 0 p trials) on peak velocity residuals, within each group (top row: PD ON, PD OFF, HC), and between medication conditions (bottom row). Dots show the mean values. No Spearman's correlations were significant (ON: ρ = −0.1549, p=0.4503; OFF: ρ = 0.3730, p=0.0614; HC: ρ = −0.2153, p=0.2609; Contingent ON vs OFF: ρ = −0.3429, p=0.0869; Guaranteed ON vs OFF: ρ = 0.1432, p=0.4834; ON-OFF Contingent vs Guaranteed: ρ = −0.2438, p=0.2291).

The online version of this article includes the following source data for figure 5:

**Source data 1.** Source individual data for velocity residual correlations for PD ON, OFF and HC.

PD OFF had more microsaccades (<1°) during the 1400ms fixation period than PD ON (F (1, 201) = 5.0451, p = .0258, $\eta_p^2$ = . 0245), but there were no other effects or interactions (p >.05, Supplementary File 3A for statistics). Conversely, ocular drift speed was higher in PD ON than OFF (F (1, 216) = 5.4327, p = .0207, $\eta_p^2$ = .0245), but there were no other significant effects or interactions (p >.05, see Supplementary File 3B). Importantly, the lack of interactions means that while patients may have differed in their fixation activity, this was unaffected by motivation conditions, and thus a different pattern to the main effects shown above.

To quantify ocular tremor, we performed Fourier transforms on the eye position in the early (200–700 ms) and late (700–1200 ms) fixation periods, and compared these between conditions with cluster-wise permutation tests to look for clusters of frequencies where patients differed. We found no significant clusters (p>0.05).

## Discussion

In this study, we tested two competing theories of dopaminergic motivation – that dopamine improves instrumental, contingent motivation, and that dopamine improves guaranteed reward motivation via reward expectation. Patients with PD made more vigorous responses, measured by peak saccade velocity residuals (*Figure 2a*), when rewards were either contingent on performance or guaranteed, but these two effects were differentially affected by dopaminergic medication. When ON medication, PD patients were motivated by rewards contingent on performance, but not by guaranteed rewards. In contrast, when patients were OFF their dopaminergic medication, the

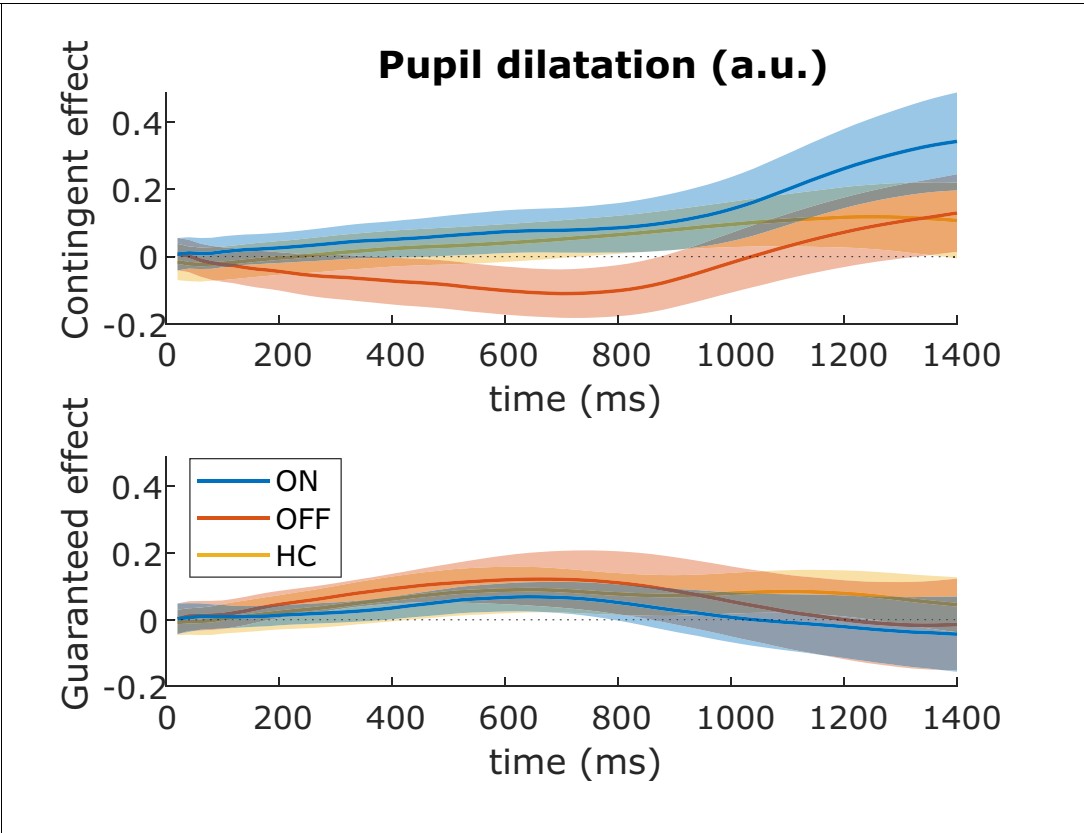

**Figure 6.** No effects of motivation on pupil dilatation. The effects of contingent (top) and guaranteed rewards (bottom) on pupil dilatation in the different conditions up to 1400 ms after the reward cue. Pupil dilatation is baselined to the time of cue onset. There were no significant clusters of difference between any groups (cluster-wise permutation testing: p>0.05), nor did a window-of-interest (1000–1400 ms) ANOVA find any significant effects (*Supplementary file 2A-C*). Shading shows SEM. Source data are available in *Figure 6—source data 1*. *Figure 6—figure supplement 1*. No correlation of pupil dilatation and motivational effects on velocity. *Figure 6—figure supplement 2*. Individual data for pupil dilatation. The online version of this article includes the following source data and figure supplement(s) for figure 6:

**Source data 1.** Source individual data for pupil dilatation for PD ON, OFF and HC.
**Figure supplement 1.** No correlation of pupil dilatation and motivational effects on velocity.
**Figure supplement 2.** Individual data for pupil dilatation.

opposite pattern was observed; they were motivated by guaranteed rewards, but not by rewards contingent on performance. In this study, older healthy controls were not significantly invigorated by either guaranteed or contingent rewards, although they showed a numerically similar pattern to PD ON. Guaranteed rewards led to PD OFF having earlier increases in velocity and acceleration (*Figure 3c & d*), which was not seen in PD ON or when rewards were contingent, and this was accompanied by increased autocorrelation of eye position (*Figure 4*), suggesting increased motor noise early in the saccade. The two motivational effects were uncorrelated across people and between medication states (*Figure 5*) indicating that dopamine does not promote one type of motivation over another in a competitive fashion, and were not associated with changes in pupil dilatation (*Figure 6*). Rather, reward expectation and contingency provide distinct motivational drives (*Figure 7*), which can be dissociated by dopaminergic medication.

The results suggest that dopamine is necessary for contingent motivation. Contingent motivation requires the use of stimulus-action-outcome associations for goal-directed behaviour (*Daw and Dayan, 2014*; *Dickinson, 1985*), while reward expectation can occur via stimulus-outcome associations (*Niv et al., 2007*) that do not require understanding the causal role of action. Our results align with rodent work demonstrating that dorsomedial striatum dopaminergic lesions impair action-outcome associations, such that animals continue to respond to previously rewarding cues even when action-contingency is removed (*Lex and Hauber, 2010b*). At a more general level, our result is also

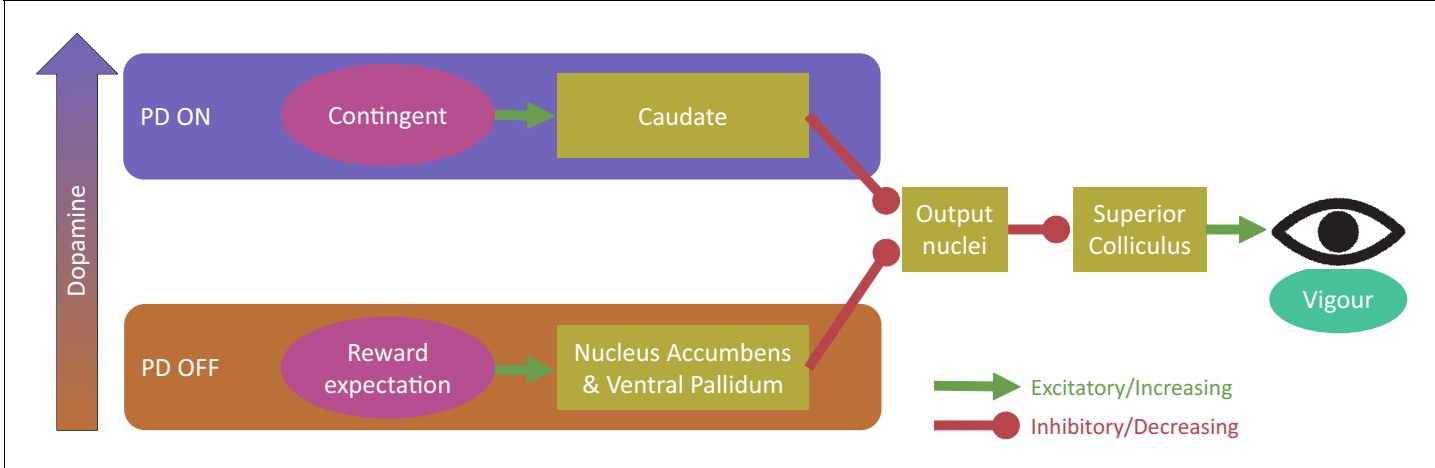

**Figure 7.** Proposed model for dopaminergic dissociation of reward expectation and contingent motivation. We propose that dopamine (in PD patients) increases contingent motivation by acting on the caudate nucleus, which disinhibits the superior colliculus (via the basal ganglia output nuclei) and affects the firing activity within the saccade, influencing vigour. Separately, high tonic dopamine impairs reward expectation motivation via the nucleus accumbens and ventral pallidum, which also disinhibit the basal ganglia output nuclei to affect superior colliculus firing activity and thus vigour within the saccade. Possible mechanisms for this dissociative dopamine influence include separate dopaminergic regions innervating the two pathways, 'global' vs 'local' signalling, or different expression of D1-like and D2-like receptors (see text for details).

consistent with dopamine being necessary for behaviours involving a causal state-action-state model (*Sharpe et al., 2017*), but not simple value-guided actions (*Sharp et al., 2016*).

Our finding reveals that dopaminergic medication attenuates the cue-driven reward expectation effect on vigour can be contrasted with previous work suggesting that tonic dopamine couples vigour to average reward rate (*Beierholm et al., 2013*; *Niv et al., 2007*). Our adaptive reward schedule held the average reward rate constant *over time*, while manipulating the average reward rate *within each condition*, such that the guaranteed 10 p and 0 p trials had different expected rewards. Dopamine might reduce these expectation effects through a different mechanism; the guaranteed cues elicit Pavlovian signals that track expected rewards across states and cues rather than time. Our result implicates dopamine in this signalling, but the direction of effect contrasts with naïve predictions. Dopamine is necessary for Pavlovian-to-Instrumental transfer (*Hall et al., 2001*; *Kelley and Delfs, 1991*; *Wyvell and Berridge, 2000*) via the nucleus accumbens. In contrast, we show that reward expectation influences vigour when dopaminergic tone is low, yet does not when dopaminergic tone is high. This aligns with the finding that slow, tonic dopaminergic activity is not related to Pavlovian-to-Instrumental transfer (*Wassum et al., 2013*). A possible explanation is that being ON dopamine led to a saturation in tonic dopamine leaving little room for phasic cue-related reward expectation signals. But if this were the case, one might expect generally higher velocities when ON, compared to PD OFF, which was not seen. Because our contingent and random conditions were matched for average reward rate, and thus opportunity cost, invigoration by contingent reward indicates a truly instrumental effect.

An alternative explanation for the discrepancy with previous research showing dopamine encodes reward rate, is that the previous studies did not fully decouple contingent and non-contingent motivation. In many studies, expected rewards were only given for successful performance (*Beierholm et al., 2013*; *Niv et al., 2007*), meaning the rewards were still contingent on performance. However, when separated, contingent motivation has larger effects on vigour than reward expectation (*Manohar et al., 2017*), and so it is possible that some previously reported effects of average reward rate on vigour were due to the greater contingency, separate from or in addition to, reward expectation. Indeed, vigour may be *reduced* by dopamine in PD, though reward sensitivity is increased (*Muhammed et al., 2016*). An additional challenge to the tonic dopamine theory of reward expectation comes from the finding that fast phasic dopaminergic responses in the nucleus accumbens encode average reward rate, but slow tonic responses do not (*Mohebi et al., 2019*). That study suggests that reward expectation signals are independent of ventral tegmental area

dopaminergic neuron firing, and may instead be due to 'local' control over nucleus accumbens core dopamine release. As dopamine is depleted in PD via dopamine-neuron death in the substantia nigra and ventral tegmental area, local dopamine release in other areas may be relatively preserved, and thus still able to influence vigour when PD patients are without dopamine.

The effect of reward-expectation on peak velocity was accompanied by greater velocity, acceleration, and autocorrelation early in the saccade for PD OFF than ON. Greater autocorrelation at this point is expected, as greater velocity increases noise (*Harris and Wolpert, 2006*; *Fitts, 1954*). However, this noise increase did not persist until the end of the saccade, as there was no increase in autocorrelation at the end of the saccade (*Figure 4*) and no greater endpoint variability (*Figure 2e*) – indeed, guaranteed rewards actually decreased endpoint variability, although this was not affected by dopamine. This offers some indirect evidence that the increased noise in this condition was attenuated via negative feedback (c.f. *Manohar et al., 2019*).

PD patients had slower saccadic RTs, and slower, smaller and more variable saccades compared to age-matched controls. The pattern of invigoration also differed from controls, who did not show significant effects of either contingent motivation or reward-expectation on speed. Instead, controls had lower motor variability when rewards were guaranteed, but no other significant motivation effects. This leads to a pattern where PD ON show contingent motivation, PD OFF show reward-expectation effects, and HC show neither. As these effects themselves are not statistically different between groups, we are limited in the conclusions that we can draw about them. Numerically, controls show a similar pattern to PD ON (*Figure 2a*), with faster velocity residuals for contingent rewards, which could suggest that dopaminergic medication is restoring healthy function, but care must be taken with this interpretation. The lack of either type of motivation in the older HC is surprising given that in healthy young adults, both contingent and guaranteed rewards increase saccade velocity (*Manohar et al., 2017*). This could suggest ageing decreases both contingent-motivation and reward-expectation, although a study directly comparing ages would be needed to conclude this.

The motivational effects reported here were not related to any pupillary responses, unlike our previous findings in young people, which may be due to both ageing and PD decreasing the influence of rewards on pupil size (*Manohar and Husain, 2015*; *Muhammed et al., 2016*). Additionally, while the two distinct motivational effects on velocity were uncorrelated within PD patients, it is possible that subgroups of patients showed different effects. For example, whether patients were on D2 agonists (*Bryce and Floresco, 2019*) or had tremor-dominant disease (*Wojtala et al., 2019*) might be relevant. However, this study was not powered to detect such differences as only six patients were taking agonists in addition to levodopa.

Considering the neuroanatomical differences between contingent motivation and reward expectation may help to explain our results. The nucleus accumbens and ventral pallidum modulate their activity by reward expectation (*Mohebi et al., 2019*; *Tachibana and Hikosaka, 2012*), while the caudate nucleus is active when rewards are contingent on behaviour (*Lex and Hauber, 2010a*; *Tricomi et al., 2004*). Both the caudate and accumbens/pallidum project to the output nuclei of the basal ganglia, allowing saccade initiation via the superior colliculus, which controls not only the direction of saccades, but also their instantaneous velocity during the movement (*Smalianchuk et al., 2018*). We propose contingent motivation and reward expectation both lead to motivational signals affecting the superior colliculus' activity controlling the velocity and acceleration of saccades, and these are differentially affected by dopamine (*Figure 7*), although we remain agnostic as to the mechanism for this difference. Possibilities include the two systems receiving input from separate regions of the dopaminergic system which are differentially depleted in PD (e.g. dopamine overdose hypothesis [*Cools, 2006*]), differences in 'global' and 'local' dopamine signals (*Mohebi et al., 2019*), or differences in D1-like and D2-like receptor expression within these systems (*Surmeier et al., 2007*; *Yetnikoff et al., 2014*). Further studies should address this question of the underlying mechanism.

We have shown that in PD, dopaminergic medication boosts motivation by contingent rewards, but reduces motivation by expected reward. Nonspecific invigoration by reward may thus be generated by a different neural system than goal-directed motivation. This suggests that dopaminergic medication may be a potential treatment for impairments in contingent motivation, but not for deficits related to reward expectation.

## Materials and methods

### Participants

Thirty PD patients were recruited from volunteer databases in the University of Oxford. They were all taking levodopa medication, and some were also taking monoamine oxidase inhibitors and/or dopamine agonists (*Table 3*). They were randomly assigned to be tested ON or OFF medication first, and withdrawn from standard release medication for 16+ hours and controlled-release medication for 24+ hours. Two patients did not complete both sessions, and two did not have enough trials that passed all the criteria (see Analysis section) so were excluded, leaving 26 patients. Thirty healthy controls (HC) were recruited from volunteer databases also, and tested once, and one HC was excluded for insufficient trials passing the criteria. We recruited 30 participants in each group as this was the sample size used in previous experiments with this task and yielded robust effects (*Manohar et al., 2017*). Sensitivity power calculations showed this would detect effect sizes above 0.46 (*Faul et al., 2009*) ($\alpha$ = 0.05, power = 0.8, sample size = 30), although as we only included 26 PD in the analysis, this effect size rose to 0.5.

All participants gave written informed consent, and ethical approval was granted by the South Central Oxford A REC (18/SC/0448).

### Procedure

The task was run in Matlab (www.mathworks.com, version 7) using the Psychophysics toolbox (*Kleiner et al., 2007*), on a Windows XP computer with a CRT monitor (1024 × 768 pixels, 40 × 30 cm, 100 Hz refresh rate) at 70 cm viewing distance. Eye movements and pupil size were recorded with Eyelink1000 at 1000 Hz.

On each trial of the task a fixation dot (0.3° radius) was presented at the centre of the screen, with two empty circles (1.1° radius) shown 9.3° to the left and right of the fixation dot. After 500 ms of fixation, a cue was given by a voice over the speaker, indicating the type of trial the participant was in:

- 'Performance' indicated that fast response times would win 10 p, while slow response times would win 0 p
- 'Random' indicated a 50% probability of 10 p or 0 p, regardless of response time
- 'Ten pence' indicated a guaranteed 10 p, regardless of response time
- 'Zero pence' indicated guaranteed 0 p, regardless of response time

A delay of 1400, 1500 or 1600 ms was given (with equal probability), after which one of the two circles turned white (50% probability of left or right) and participants had to saccade to this circle to complete the trial and receive the outcome.

Participants could only affect the outcome in the Performance condition (by moving faster); all others were independent of their speed. In the Performance condition, rewards were based upon response time (i.e. total time between the target appearing and gaze arriving at the target), which is only minimally influenced by saccade velocity. Participants were rewarded when response time was quicker than their recent median response time for the last 20 Performance trials, which thus yielded a 50% reward rate overall. The Random condition acts as a control to these trials, with a random 50% of trials rewarded, and thus equal expected value but with no performance-contingency. Rewards in the guaranteed conditions also had zero contingency on performance, but yielded different expected rewards (10 p vs 0 p), thus comparing them indexes the pure effect of expecting reward.

When rewards are contingent, people get feedback about how they performed. This itself is known to increase motivation, independent of reward – a phenomenon termed intrinsic motivation. To control for this, we ensured participants always received feedback on their speed (fast/slow, using median split over 20 previous trials in that condition – i.e. the same criteria as for contingent rewards), regardless of reward. This should equate the level of intrinsic motivation across conditions, providing that the feedback is as noticeable as the reward. In order to ensure the speed feedback and reward were matched in physical salience, the feedback modalities were counterbalanced. Two blocks gave auditory feedback for speed and visual feedback for reward, and vice versa for the other two blocks, with order randomised across participants. This counterbalancing accords with our previous study (*Manohar et al., 2017*) which found no modality effects, suggesting participants were attending to audio and visual feedback equally. We also found no effects of modality on any of the measures of interest (p>0.05), so collapsed across the two modalities for all the analyses.

There were 12 types of each trial in a block, in a random order, and participants completed four blocks.

## Analysis

The Performance and 10 p conditions are high motivation conditions. The difference between Performance and Random conditions gives the effect of contingent motivation, while the difference between 10 p and 0 p conditions gives the effect of reward expectation.

As in previous studies (*Manohar et al., 2017*), our primary measure of interest was saccadic vigour. We measured peak saccade velocity on each trial. We took the first saccade after target onset which was greater than 1° in amplitude, and used a sliding window of 4 ms width to calculate velocity, excluding segments faster than $3000°s^{-1}$ or where eye tracking was lost. Saccades with peak velocities outside $80–2500°s^{-1}$ were excluded, as were trials where participants reached the target before 180 ms or after 580 ms. Two PD patients and one HC had fewer than 10 trials that passed these criteria for one condition, so were excluded from the analysis.

To remove the main sequence effect of amplitude on velocity (*Bahill et al., 1975*; *Harris and Wolpert, 2006*), we regressed velocity against amplitude and took the peak velocity residuals as our measure of interest. This measures the difference between the velocity predicted by the main sequence, and the velocity actually recorded, with positive (negative) values meaning faster (slower) velocity. This was done for each participant's separate session. This approach has been used before, by us and others (*Blundell et al., 2018*; *Manohar et al., 2017*; *Muhammed et al., 2020*; *Muhammed et al., 2016*; *Van Opstal et al., 1990*), and it is similar to simply including amplitude as a covariate when analysing raw peak velocity, but it does not reduce the degrees of freedom and yields simpler to interpret results. Moreover since motivation increases amplitude (*Manohar et al., 2019*), including amplitude as a covariate would mean that amplitude would compete with motivation to explain variance in velocity, potentially resulting in overestimation of motivation effects. Our findings did not qualitatively change when we used the covariate approach instead.

We also measured amplitude, saccadic reaction time (RT), and endpoint variability of these saccades. Saccadic RT is the time between the target onset and the start of the saccade.

To analyse velocity and acceleration traces, and autocorrelation and covariance of the eye movements we linearly interpolated 50 points along each saccade to move them into the same units. Instantaneous velocity was smoothed across three time-points, while acceleration was smoothed across 5. We also calculated velocity and acceleration traces on the raw (non-interpolated) traces and then interpolated them afterwards, which gave very similar results.

Pupil dilatation was measured in arbitrary units (a.u.) relative to the baseline pupil size at the cue onset. Blinks under 500 ms were linearly interpolated, steps in pupil size above 2.5 a.u./ms were removed, and data were averaged in 20 ms bins for plotting.

We used rmanova from the matlib toolbox (https://github.com/sgmanohar/matlib; *Manohar, 2020*) to perform analyses – this uses fitglme to perform the repeated-measures test and anova to perform hypothesis tests on the GLME. We used three-way repeated-measures ANOVA to compare effects of motivation, contingency and dopaminergic medication in PD patients, and followed this up with two-way ANOVA when a three-way interaction was found. These analyses were also performed using a full linear mixed effects model including each trial, which produced qualitatively identical results. To compare each PD condition against HC we used mixed ANOVA. We also used cluster-wise permutation tests for the time-course data (velocity, acceleration, pupil dilatation, autocorrelation and covariance), to control the family-wise error rate at. 05.

## Data and code availability

Analyses were performed in Matlab using custom scripts, which are available on GitHub (https://doi.org/10.5281/zenodo.4032711). Anonymous data are available on OSF (https://osf.io/2k6x3), as is the experiment file (osf.io/y9xhp) https://doi.org/10.5281/zenodo.4032711.

## Acknowledgements

We would like to thank the patients and participants for their time in taking part in this study, and the funders for their support (MRC Clinician Scientist Fellowship to SGM, MR/P00878X).

## Additional information

### Competing interests
Michele T Hu: MTH is a consultant advisor to the Roche Prodromal Advisory, Biogen Digital Advisory Board, Evidera, and CuraSen Therapeutics, Inc. The other authors declare that no competing interests exist.

### Funding

| Funder | Grant reference number | Author |
|--------|------------------------|--------|
| MRC | MR/P00878X | Sanjay G Manohar |

The funders had no role in study design, data collection and interpretation, or the decision to submit the work for publication.

### Author contributions
John P Grogan, Resources, Data curation, Software, Formal analysis, Investigation, Visualization, Methodology, Writing - original draft, Project administration, Writing - review and editing; Timothy R Sandhu, Data curation, Investigation, Methodology, Writing - review and editing; Michele T Hu, Resources, Writing - review and editing; Sanjay G Manohar, Conceptualization, Resources, Supervision, Funding acquisition, Methodology, Writing - review and editing

### Author ORCIDs
John P Grogan (iD) https://orcid.org/0000-0002-0463-8904
Sanjay G Manohar (iD) http://orcid.org/0000-0003-0735-4349

### Ethics
Human subjects: Ethical approval was granted by the South Central Oxford A REC (18/SC/0448). All participants gave written informed consent.

### Decision letter and Author response
Decision letter https://doi.org/10.7554/eLife.58321.sa1
Author response https://doi.org/10.7554/eLife.58321.sa2

## Additional files

### Supplementary files
- Supplementary file 1. Statistics for main saccade measures.
- Supplementary file 2. Statistics for Pupil Dilatation.
- Supplementary file 3. Statistics for fixation period.
- Supplementary file 4. Correlations of questionnaires with motivational effects.
- Transparent reporting form

### Data availability
Anonymised data are available on OSF (https://osf.io/2k6x3).

The following dataset was generated:

| Author(s) | Year | Dataset title | Dataset URL | Database and Identifier |
|-----------|------|---------------|-------------|-------------------------|
| Grogan JP, Sandhu TR, Hu MT, Manohar SG | 2020 | Contingent Motivation in PD | https://osf.io/2k6x3 | Open Science Framework, 2k6x3 |

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
