## [Decision Letter]

**Acceptance summary:**

The manuscript describes a unique and elegantly designed experiment to parse the role of dopamine in reward learning using human subjects – namely, patients with Parkinson's Disease, on and off dopaminergic medication. The results are novel, showing that patients with Parkinson's Disease that were ON medication had greater response vigor for contingent rewards; while those OFF medication had greater vigor for guaranteed rewards. These findings support the long-standing notion that reward expectation and contingency learning represent distinct motivational and neurobiological processes.

**Decision letter after peer review:**

Thank you for submitting your article "Dopamine promotes instrumental motivation, but reduces reward-related vigour" for consideration by *eLife*. Your article has been reviewed by two peer reviewers, including Shelly B Flagel as the Reviewing Editor and Reviewer #1, and the evaluation has been overseen by Michael Frank as the Senior Editor. The following individual involved in review of your submission has agreed to reveal their identity: Roshan Cools (Reviewer #2).

The reviewers have discussed the reviews with one another and the Reviewing Editor has drafted this decision to help you prepare a revised submission.

Summary:

This is a timely paper on a topic of great interest with intriguing findings that incidentally also reconcile some paradoxical findings in extant literature. A clever novel rewarded saccade task was used to disentangle effects of dopaminergic medication in Parkinson's disease on two distinct types of motivation: contingent (instrumental) motivation, where reward depends on performance versus non-contingent (non-instrumental) motivation, where reward is guaranteed. Results reveal a unique double dissociation, with contingent motivation (indexed by peak velocity) being greater ON than OFF meds, but noncontingent motivation being greater OFF than ON meds. Thus PD patients OFF their medication showed a positive effect of non-contingent reward on vigor, but no effect of contingency, whereas the same patients ON their dopaminergic medication showed a positive effect of contingent vs non-contingent reward on vigor, but no effect of non-contingent reward.

This is a robust set of findings and a beautiful cross-over effect, not often seen, obtained using a well-controlled paradigm. The authors provide thorough auxiliary and control analyses to validate the specificity of their effect of interest.

Revisions:

Some concerns raised by both reviewers, regarding data analysis, interpretation and conclusions are outlined below. Of particular note, the authors need to better incorporate the reported findings into the extant literature.

1) Given the involvement of dopamine in "expectation", did the authors investigate any potential impact of order effects? That is, if the subjects were tested ON or OFF medication first, did that impact subsequent responding? It may be that the sample size is too low for this analysis and/or for the correlations to be meaningful; nonetheless, this possibility should be discussed.

2) As peak saccade velocity residuals were the primary outcome measure affected, this metric warrants further description and justification in the primary text, prior to the results.

3) It seems that HC subjects are not included in the primary analyses, but only compared to PD ON and PD OFF, in separate analyses, in the supplemental text. HC subjects should be included in the primary analyses, as they are shown in Figure 2. In relation, the authors should further interpret the fact that HC subjects differ from PD patients on measures of amplitude, saccadic RT and endpoint variability, but seem to resemble PD ON for peak velocity residuals (as shown in Figure 2). While the authors briefly discuss the negative results in HC subjects, it is not clear why there is no effect of motivation in these subjects, as one might expect. The pattern observed in HC subjects might help to explain the data from the PD subjects and therefore should be better incorporated into the manuscript. HC data should be included in Figure 3 and the accompanying analyses, as well.

4) Additional "discussion should be provided regarding the conclusion that these data suggest that dopamine is necessary only for model-based, and not model-free learning; as there are other potential explanations; especially since the HC data don't seem to support model-based learning in this task.

5) Considerable parts of the Introduction and Discussion should be rewritten. Specifically, the data do not undermine the (i) theory that dopamine is key for reward expectation (Abstract: “challenging the theory that tonic dopamine encodes reward expectation”) or (ii) average reward rate (Discussion: “this contrasts with previous research which suggests tonic dopamine encode the average reward rate”) or (iii) model-free behavior (“thus the work fits with previous research in which dopamine is necessary only for model-based, not model-free learning”). These conclusions are not supported by the current findings, because:

i) Detrimental effects of dopamine implicate dopamine as much as do beneficial effects as the authors conclude later on in the discussion.

ii) Average reward rate was not manipulated and its effects not measured. Related to this issue, I do not think that the Niv view is optimally characterized by attributing to it the prediction that dopamine would leave contingent motivation unaltered.

iii) The noncontingent Pavlovian procedure tested here doesn't address model-free learning, as measured in the papers referenced. Moreover, evidence for impaired model-based vs intact model-free control in PD (first evidenced by de Wit et al. in JoCN) is not directly relevant here, given the fairly wide gap between the type of inference that is measured in e.g. the two-step task and similar other learning tasks and the contingent reward motivation manipulation measured here.

Some of these conclusions (e.g. (i)) also counter the hypothesis put forward later in the discussion and illustrated in Figure 6 about the two forms of motivation implicating distinct dopaminergic mechanisms.

6) Two sets of literature that are directly relevant to this research are absent from the manuscript and need to be incorporated.

• The first is (pharmacological and lesion) evidence from work with experimental rodents, but also humans, on dopamine's role in the Pavlovian control of behavior, e.g. on conditioned reinforcement (Parkinson et al; Taylor and Robbins), Pavlovian-to-instrumental transfer (Dickinson et al., 2001; Wyvell and Berridge, 2000; Talmi et al., 2008), autoshaping, goal vs sign tracking etcetera.

• The second is prior evidence for enhanced effects of reward motivation in PD patients OFF but not ON meds in a manner that depends on dopamine cell loss (Aarts et al., 2012, and Timmer et al., 2018). The Introduction should reflect this prior evidence. In fact, the rationale for the current study is strengthened by the apparent discrepancy between this evidence for enhanced motivation in PD OFF meds (somewhat reminiscent of paradoxical kinesia) and the evidence previously obtained by the authors for reduced motivation in PD OFF meds.

7) The current version of the Introduction (as well as some of the Discussion) is structured based a contrast between model-based versus Pavlovian control of behavior that does not do justice our understanding of these concepts from classic learning theory. For example, the statement that “Motivation by reward expectation, independent of what an agent does, is model-free” should be rephrased. Pavlovian control can be model-free or model-based (Dayan and Berridge, 2014, CABN or Bornstein and Daw, 2011, Curr Opin Neurobiol). Related to this is my issue with the conclusion in the Discussion (see above).

8) Some of the statistical choices need to be justified and, data presentation optimized.

• The use of peak velocity residuals, corrected for amplitude, should be justified more clearly.

• It would also be good to clarify the rationale for using these residuals rather than correcting the velocities for amplitude with a covariate of no interest e.g. in a trial-wise mixed linear regression analysis.

• The use of mixed linear modeling would allow both between- as well as within-subject variability to be taken into account. Was this considered?

• I suggest that readers get access to uncorrected, observed velocity data rather than only the residuals.

• While individual datapoints are presented in the supplement, I see no principled reason to present aggregate dataplots in the main text. Why not move the supplementary figures with actual datapoints to the main text? Raincloud plots might increase the readability of these plots?

---

## [Author Response]

Revisions:Some concerns raised by both reviewers, regarding data analysis, interpretation and conclusions are outlined below. Of particular note, the authors need to better incorporate the reported findings into the extant literature.

We thank the reviewers for their time and their helpful comments. We have addressed all of them, and believe it has made the manuscript more robust and readable. We address each of the points below.

When adding individual data points and HC data to some figures, we had to split some figures into new separate ones. We also noticed that some participants had not been excluded from the autocorrelation and pupillometry data, so have redone those analyses to match the exclusions applied to the other data. This has resulted in a new cluster of significance in PD OFF which matches up with the previous ON vs OFF difference, so has not changed our findings. We also corrected a typo in the demographics Table 3 (PD ACE was wrong), and have moved legends and tables to the end of this file and uploaded figures separately as requested by *eLife* editorial support.

1) Given the involvement of dopamine in "expectation", did the authors investigate any potential impact of order effects? That is, if the subjects were tested ON or OFF medication first, did that impact subsequent responding? It may be that the sample size is too low for this analysis and/or for the correlations to be meaningful; nonetheless, this possibility should be discussed.

We agree with the reviewer that order effects could be important. Including which session (ON/OFF) was performed first as a factor in the analysis revealed no effects of session order, or any interactions with session order, on any of the saccade measures reported. We have stated this in the Results section:

“We additionally checked whether there were practice effects in the PD patients, in case patients behaved differently on their second session due to different expectations. We found no effects or interactions of session on any measure in PD patients (p >.05).”

2) As peak saccade velocity residuals were the primary outcome measure affected, this metric warrants further description and justification in the primary text, prior to the results.

We have further explained this measure:

“A saccade’s velocity is tightly governed by its amplitude, a relation known as the “main sequence” (Bahill, Clark and Stark, 1975). […] We did this regression for each participant and session separately, but across conditions.”

We have also added a panel to Figure 1 to have an illustrative example of residuals using sample data, and have updated the legend to:

“Example of the main sequence and velocity residuals – the points show a subset of individual trials illustrating the “main sequence” relationship where larger saccades have greater velocity, shown by the regression line. The distance from each point to its line is the velocity residual, which we take as out main measure of response vigour.”

3) It seems that HC subjects are not included in the primary analyses, but only compared to PD ON and PD OFF, in separate analyses, in the supplemental text. HC subjects should be included in the primary analyses, as they are shown in Figure 2. In relation, the authors should further interpret the fact that HC subjects differ from PD patients on measures of amplitude, saccadic RT and endpoint variability, but seem to resemble PD ON for peak velocity residuals (as shown in Figure 2). While the authors briefly discuss the negative results in HC subjects, it is not clear why there is no effect of motivation in these subjects, as one might expect. The pattern observed in HC subjects might help to explain the data from the PD subjects and therefore should be better incorporated into the manuscript. HC data should be included in Figure 3 and the accompanying analyses, as well.

We agree with the reviewers that it would be useful to add interpretation to the HC results. We have therefore moved the paragraphs describing the HC-only findings to earlier in the first Results section, and rewritten it to be clearer. We have explained why residual velocity did not show a main effect of PD vs HC, and added a sentence evaluating the comparison to HC in the Results section:

“The HC peak velocity residuals were not affected by contingency, motivation or the interaction (p >.05; see Table 2), suggesting that healthy older adults do not adjust their response vigour for contingent or guaranteed rewards. […] HC did not significantly differ from PD ON or OFF in peak velocity residuals, although their pattern was numerically closest to PD ON with greater contingent motivation.”

And have expanded the Discussion of these results:

“PD patients had slower saccadic RTs, and slower, smaller and more variable saccades compared to age-matched controls. […]This could suggest ageing decreases both contingent motivation and reward-expectation, although a study directly comparing ages would be needed to conclude this.”

We have added HC data to all data figures, and the accompanying analyses. We have moved the autocorrelation data to a new separate figure in order to include the HC data, and we now present two correlation matrices (for contingent and guaranteed reward effects) for each group (PD ON, OFF, HC) and the ON-OFF difference.

4) Additional "discussion should be provided regarding the conclusion that these data suggest that dopamine is necessary only for model-based, and not model-free learning; as there are other potential explanations; especially since the HC data don't seem to support model-based learning in this task.

We have rewritten this part of the discussion, discussing PIT and instrumental model-free associations, and have clarified that model-based associations are only one possibility here:

“The results suggest that dopamine is necessary for contingent motivation. […] At a more general level, our result is also consistent with dopamine being necessary for behaviours involving a causal state-action-state model (Sharpe et al., 2017), but not simple value-guided actions (Sharp et al., 2016).”

5) Considerable parts of the Introduction and Discussion should be rewritten. Specifically, the data do not undermine the (i) theory that dopamine is key for reward expectation (Abstract: “challenging the theory that tonic dopamine encodes reward expectation”) or (ii) average reward rate (Discussion: “this contrasts with previous research which suggests tonic dopamine encode the average reward rate”) or (iii) model-free behavior (“thus the work fits with previous research in which dopamine is necessary only for model-based, not model-free learning”). These conclusions are not supported by the current findings, because:i) Detrimental effects of dopamine implicate dopamine as much as do beneficial effects as the authors conclude later on in the discussion.

We have thought carefully about the interpretation of our findings, and we agree with the reviewers that our main result does not fit exactly where we placed it relative to the extensive animal literature.

We have changed this claim:

“We posit that dopamine promotes goal-directed motivation, but dampens reward-driven vigour, contradictory to the prediction that increased tonic dopamine amplifies reward expectation.”

ii) Average reward rate was not manipulated and its effects not measured. Related to this issue, I do not think that the Niv view is optimally characterized by attributing to it the prediction that dopamine would leave contingent motivation unaltered.

The reviewer correctly argues that although we manipulated the average reward rates between conditions, the reward rate *over time* was not manipulated. It is therefore not possible for us to directly address the Niv view. We have clarified this and added a sentence pointing out this difference may be relevant for the lack of effect. We also mention that the fact dopamine reduces guaranteed motivation is in fact consistent with dopamine’s being involved with this kind of motivation:

“Our finding that dopaminergic medication attenuates the cue-driven reward expectation effect on vigour can be contrasted with previous work suggesting that tonic dopamine couples vigour to average reward rate (Beierholm et al., 2013; Niv et al., 2007). […] Because our contingent and random conditions were matched for average reward rate, and thus opportunity cost, invigoration by contingent reward indicates a truly instrumental effect.”

iii) The noncontingent Pavlovian procedure tested here doesn't address model-free learning, as measured in the papers referenced. Moreover, evidence for impaired model-based vs intact model-free control in PD (first evidenced by de Wit et al. in JoCN) is not directly relevant here, given the fairly wide gap between the type of inference that is measured in e.g. the two-step task and similar other learning tasks and the contingent reward motivation manipulation measured here.

We have shifted our focus away from “model-based vs model-free” and acknowledge that our task does not help understand the traditional distinction between forward planning vs reinforcement learning. We now clarify that the task is probing motivation through action-outcome knowledge, and is related to Pavlovian vs instrumental drives (see response to point 4 for pasted text).

Some of these conclusions (e.g. (i)) also counter the hypothesis put forward later in the discussion and illustrated in Figure 6 about the two forms of motivation implicating distinct dopaminergic mechanisms.

We have adjusted these claims, and believe the new literature added to the discussion fits with the proposal of two separate systems.

6) Two sets of literature that are directly relevant to this research are absent from the manuscript and need to be incorporated.• The first is (pharmacological and lesion) evidence from work with experimental rodents, but also humans, on dopamine's role in the Pavlovian control of behavior, e.g. on conditioned reinforcement (Parkinson et al; Taylor and Robbins), Pavlovian-to-instrumental transfer (Dickinson et al., 2001; Wyvell and Berridge, 2000; Talmi et al., 2008), autoshaping, goal vs sign tracking etcetera.• The second is prior evidence for enhanced effects of reward motivation in PD patients OFF but not ON meds in a manner that depends on dopamine cell loss (Aarts et al., 2012 and Timmer et al., 2018). The Introduction should reflect this prior evidence. In fact, the rationale for the current study is strengthened by the apparent discrepancy between this evidence for enhanced motivation in PD OFF meds (somewhat reminiscent of paradoxical kinesia) and the evidence previously obtained by the authors for reduced motivation in PD OFF meds.

We thank the reviewers for pointing out these highly relevant bodies of literature. We have rewritten the Introduction to reflect this, and added them to the Discussion (see above):

“Contingent rewards motivate us because we understand the causal relation between successful actions and reward. […] We tested the two predictions that dopamine is involved in motivation by expected rewards, and by contingent rewards.”

7) The current version of the Introduction (as well as some of the discussion) is structured based a contrast between model-based versus Pavlovian control of behavior that does not do justice our understanding of these concepts from classic learning theory. For example, the statement that “Motivation by reward expectation, independent of what an agent does, is model-free” should be rephrased. Pavlovian control can be model-free or model-based (Dayan and Berridge, 2014, CABN or Bornstein and Daw, 2011, Curr Opin Neurobiol). Related to this is my issue with the conclusion in the Discussion (see above).

We thank the reviewer for advising on these very relevant distinctions, which we now make in our narrative. We have rewritten the Introduction and the discussion to modify these claims. We had previously attempted to suggest that model-based reasoning could underlie the PD ON benefit, although we were not clear enough that this was only one possibility. We have refocused the paper on the more clear distinction between goal-directed action-outcome and Pavlovian stimulus-outcome associations (see responses above).

8) Some of the statistical choices need to be justified and, data presentation optimized.• The use of peak velocity residuals, corrected for amplitude, should be justified more clearly.

We have justified this (please see response to point 2 above).

• It would also be good to clarify the rationale for using these residuals rather than correcting the velocities for amplitude with a covariate of no interest e.g. in a trial-wise mixed linear regression analysis.

We have added the following to the Materials and methods section, justifying our use of residuals in ANOVA:

“To remove the main sequence effect of amplitude on velocity (Bahill et al., 1975; Harris and Wolpert, 2006), we regressed velocity against amplitude and took the peak velocity residuals as our measure of interest. […] Our findings did not qualitatively change when we used the covariate approach instead.”

• The use of mixed linear modeling would allow both between- as well as within-subject variability to be taken into account. Was this considered?

We planned in advance to use repeated measures ANOVA (i.e. a mixed linear model on the condition means). But as the reviewer suggests, trial-wise mixed models can provide a much more sensitive test and account for the different numbers of valid trials between conditions and subjects. We performed this analysis. For velocity residuals we obtained the following results shown in Author response table 1:

**Author response table 1. resptable1:** 

	Effect	Estimate	t	p
Peak Velocity Residuals	Motivation	3.0488	1.922	0.0546
	Contingency	0.1392	0.0878	0.9301
	Drug	-0.0539	-0.0339	0.9729
	Motivation * Contingency	0.2380	0.1500	0.8808
	Motivation * Drug	0.5420	0.3417	0.7326
	Contingency * Drug	2.6239	1.6542	0.0981
	Contingency * Motivation * Drug	3.5318	2.2266	*0.0260

In fact, for each variable, the pattern of significant effects turned out the same for both methods. Thus we added the following to the Materials and methods/analysis section:

“All statistical analyses were also performed using a full linear mixed effects model including each trial, which produced in qualitatively identical results.”

• I suggest that readers get access to uncorrected, observed velocity data rather than only the residuals.

We have included raw velocity as an extra panel in Figure 2, included the statistics in all main and supplementary results tables, and discussed this measure in the Results:

“Raw peak velocity had an effect of motivation, as both types of motivation increased speed (Figure 2F, p=0.0110), although this will include effects of changes in amplitude (via the main sequence) which showed a borderline significant effect of motivation (Figure 2C, p=0.0607).”

• While individual datapoints are presented in the supplement, I see no principled reason to present aggregate dataplots in the main text. Why not move the supplementary figures with actual datapoints to the main text? Raincloud plots might increase the readability of these plots?

We have added individual data points to the line graphs in Figure 2 (raincloud plots were very busy due to the 2*2*3 design). A few HC had strong motivation effects, which expanded the y-axes a lot, making the effects harder to see (see Author response image 1 for example),

**Author response image 1. sa2fig1:** 

So we have put the HC data on a separate panel (Figure 2B) to the PD data (Figure 2A) which we hope will be acceptable.The correlation plots already show individual data, but we were unable to include individual participants’ timecourses of velocity, acceleration, autocorrelation, or pupillometry on the respective figures as they swamped the means. Instead, we have added figure supplements showing these individual data, which we hope will be acceptable.

Figure 3—figure supplement 1 and Figure 6—figure supplement 2 are the new supplementary figures showing individual velocity and acceleration data, and pupil dilatation.

While making the individual pupillometry and autocorrelation plots, we realised we had not excluded the 4 PD and 1 HC, as we had for the main analyses. We have now updated our analyses and figures to have 26 PD and 29 HC to match the main saccade data. This affects Figures 4 and 6 only, and has not changed the findings except that PD OFF now have a significant cluster of higher autocorrelation at the beginning of the saccade for guaranteed motivation (Figure 4).

This increase when off (panel g) now matches the significant effect of dopamine (panel e) found when comparing PD ON vs OFF autocorrelations, so has not changed our interpretation. We have added:

“This coincides with the greater acceleration PD OFF patients had at the beginning of saccades to guaranteed rewards (Figure 3D), as faster movements have greater motor noise (Harris and Wolpert, 1998, 2006).”

The pupil size text has not changed.